# Transfusion with Blood Plasma from Young Mice Affects rTg4510 Transgenic Tau Mice Modeling of Alzheimer’s Disease

**DOI:** 10.3390/brainsci13060841

**Published:** 2023-05-23

**Authors:** Carlos M. Hernandez, Rachel E. Barkey, Kristen M. Craven, Karin A. Pedemonte, Bernadette Alisantosa, Jonathan O. Sanchez, Jane M. Flinn

**Affiliations:** Department of Cognitive and Behavioral Neuroscience, George Mason University, Fairfax, VA 22030, USA; chernan8@masonlive.gmu.edu (C.M.H.); rbarkey@gmu.edu (R.E.B.); kcraven2@masonlive.gmu.edu (K.M.C.); karin88@vt.edu (K.A.P.); bernadette.alisantosa@nih.gov (B.A.); jsanche@masonlive.gmu.edu (J.O.S.)

**Keywords:** neurofibrillary tangles, blood transfusion, blood plasma, Alzheimer’s disease

## Abstract

Alzheimer’s disease (AD) is characterized by the buildup of plaques and tangles in the brain. Tangles are formed when the stabilizing protein, tau, becomes hyperphosphorylated and clumps together. There are limited treatments for AD; therefore, the exploration of new treatments is warranted. Previous research showed that plasma transfusion from young donor mice improved spatial memory and increased synaptic proteins in old transgenic APP/PS1 mice, suggesting a remediation of memory and synaptic function. In the current study, plasma was transfused from 2–3-month-old young wildtype mice (WT) to 8-month-old rTg4510 mice expressing human tau (Tau). One week after the transfusions, behavior and tau pathology were examined. We found that Tau mice injected with plasma had lower expression of phosphorylated tau (ptau) in the brain, accompanied by fewer tau tangles in the cortex and CA1 region of the hippocampus and smaller tau tangles in the cortex, when compared to Tau mice injected with saline. Despite no improvement in behavior, the decreased level of ptau and tangles open the door to future studies involving plasma transfusions.

## 1. Introduction

One of the major pathologies of Alzheimer’s disease (AD) is neurofibrillary tangles (NFT), which is when tau accumulates around cellular bodies [1,2]. In a healthy brain, tau proteins stabilize microtubules, allowing neuronal connections to develop and remain intact, which is vital for learning and memory [3]. In AD, however, microtubule-associated protein tau (MAPT) undergoes hyperphosphorylation and detaches from the microtubule (Rawat). When this hyperphosphorylation occurs, it often leads to aggregation of tau, which in turn creates NFTs [4]. NFTs disrupt normal neuron communication, cellular apoptosis, and neuronal loss throughout the brain, which lead to the cognitive impairment seen in AD patients [5]. AD patients also suffer from spatial memory, which stems from damaged hippocampal and entorhinal cortex caused by NFTs [6]. In the hippocampus, hippocampal pyramidal neurons lose a significant number of dendritic spines, thus compromising learning and memory [7]. Typically, with AD, formation of NFTs starts in the entorhinal cortex, which spreads to the surrounding of the limbic system and then progresses to the hippocampus and eventually the neocortex [8]. Drug trials for AD have failed for multiple potential reasons, including limited specificity of the enzyme or receptor used and the timing of treatment initiation [9]. Recent AD treatments have focused on inhibiting tau hyperphosphorylation by slowing the progression of tauopathies, but we still do not fully understand the mechanism behind tauopathies and its toxicity on neurological functions [8]. Due to the lack of success of current treatments, the exploration of innovative treatments and novel means of administration are warranted, including by focusing on tau and, at the same time, on a systemic level.

Recently, blood plasma transfusions have received attention as an alternative to reversing the effects of aging and as a potential for treating AD. Young mice contain rejuvenating blood protein factors that can increase plasticity, dendritic spine density, neurogenesis, and improve learning and memory when intravenously injected into aged mice [10,11]. Middeldorp et al. (2016) injected 10–12 month-old APP mice with blood plasma from 2–3-month-old wildtype mice [12]. The APP mice treated with plasma had improved hippocampal-dependent memory. Young plasma also restored phosphorylated mitogen-active extracellular signal-regulated kinases (pERK/ERK), synaptophysin levels, and calbindin level to the levels of the young control mice but did not affect amyloid plaques in the APP mice. This study sought to see whether plasma affected other pathologies commonly seen in AD, such as NFTs.

The objective of the current study was to assess the effects of young plasma on NFTs. The only way to investigate tangles alone is with the use of a non-APP mouse model. This was accomplished by using the rTg4510 animal model that expresses human tau (Tau) and develops NFTs in the hippocampus and cortex. The rTg4510 animal model is ideal for this study as it expresses mutant human tau at 13 times higher than mouse tau, develops NFTs and neuronal death, and exhibits behavioral impairments [13]. At 2.5 months, pre-tangles and gliosis reactivity, based on GFAP detection, is observed [14]. By 5.5 months, NFTs are observed in the hippocampus [13,15]. By 9.5 months, dendritic spines diminish, further impairing memory and cognition due to the shortage of presynaptic milieu [16].

Plasma administration did reduce the number of tangles and phosphorylated tau in the transgenic (Tau) mice but did not improve behavioral performance. Nevertheless, this study increases our knowledge on how young plasma may affect NFTs and paves the way for further experiments, possibly with younger animals that are not yet irreversibly affected by the expression of Tau.

## 2. Materials and Methods

The study described in this paper was approved by the George Mason University Institutional Animal Care and Use Committee under protocol #0390, entitled “Effects of Blood Plasma Transfusion from Young Mice to Aged h-tau Mice Modelling Alzheimer’s Disease”. The IRBNet package number was 1195958-1. The procedures described herein were in accordance with the AVMA and The Guide for the Care and Use of Laboratory Animals, as well as the AAALAC guidelines.

### 2.1. Breeding and Housing

Male CaMKII-tTA (JAX #007004) mice were paired with female rTg4510 (JAX #015815) mice in harems of 1 male to 3 females to breed the experimental Tau P301L rTg4510 (CAMKII:hTau-P301L) “Tau” mice, which develop NFTs in the forebrain. The mice were fed Love Mash (BioServ: S3823P) for 2 weeks before pairing. Tail snips were collected before 21 days for genotyping (Transnetyx (Cordova)). The mice were weaned and placed in standard Animal Care System (Centennial) rat cages with a Mouse Igloo and Fast Trac (BioServ) for enrichment, and housed by sex and with littermates, while being given Teklad 7012 feed (Envigo) ad libitum and access to water through a lixit system (Edstrom); the rooms were on a 12 h light/12 h dark cycle.

### 2.2. Donor Animals and Blood Collection

Blood was collected via a terminal intracardiac bleed from FVB inbred mice (JAX #001800) between the ages of 8 and 10 weeks. The donor mice were anesthetized with 3–4% isoflurane admixed with 1–1.5 L/min of O_2_ delivered by a precision vaporizer into an induction chamber. Once deeply anesthetized, a nose cone was used to maintain 1.5–2% isoflurane admixed with 1–1.5 L/min of O_2_. Blood was extracted using an ethylenediaminetetraacetic acid (EDTA) (Sigma-Aldrich, St. Louis, MO, USA)-coated syringe with a 25 G × 5/8” needle and transferred to a centrifuge tube coated with EDTA.

### 2.3. Blood Plasma

The blood samples were centrifuged at 1000 g for 10 min at 4 °C and stored at −80 °C. Prior to administration, EDTA was removed using 3.5 kDa Slide-A-Lyzer Dialysis Cassettes (ThermoFisher Scientific, Waltham, MA, USA: 66110); the samples were washed four times for two hours in PBS and washed again overnight at 4 °C. The mice in the experimental group received 150 µL of plasma, while the mice in the control group received 150 µL of 0.9% saline. The mice were placed under a heat lamp to dilate the tail vein and then in a Broome-style rodent restrainer (Plas Labs, Lansing, MI, USA). The mice were injected 8 times in 48 h intervals with a 27 G × ½” needle, for a total transfusion of 1.2 mL.

### 2.4. Experimental Animals

Injections began at eight months of age. There were four groups: Tau mice given (a) plasma (n = 15) or (b) saline (n = 14), and wildtype mice given (c) plasma (n = 13) or (d) saline (n = 16) (Table 1). Behavioral testing began a week after all eight injections were administered, which was about three weeks after their first dose. The mice were then euthanized in accordance with the AVMA guidelines at 9½ months of age.

### 2.5. Behavioral Tests

Behavioral tests were performed in the following order (Figure 1):

#### 2.5.1. Nesting

The nesting test followed the protocol published by Neely et al. in 2019 [17].

#### 2.5.2. Burrowing

The mice were placed into individual cages. A hollow tube with one end closed was filled with 250 g of pea shingles. The remaining pea shingles in the tubes were weighed after 2 h and 12 h.

#### 2.5.3. Open Field

The open field test was used to identify anxiety and motor deficits by measuring the time spent in the center versus the surrounding area. The distance traveled and speed were recorded over a 5 min period using the TopScan (Clever Sys, Reston, VA, USA) version 1.0 software.

#### 2.5.4. Morris Water Maze

The mice were tested in a Morris water maze (MWM), following the protocol in Craven et al. (2018) (Table 2) [18]. Testing took place over eight consecutive days with 3 trials per day, except for days 7 and 8. The probe trials took place on days 2, 4, 6, and 7. Day 7 consisted of a 24 h interval probe trial. Day 8 had 2 trials using a visible platform to ensure there were no visual-deficient animals.

The distance traveled, the latency to find the platform, and the time spent in the target quadrant, and thigmotaxis to assess anxiety were measured. The recording and analyses were completed using WaterMaze3 (Coulbourn Instruments, Allentown, PA, USA).

#### 2.5.5. Circadian Rhythm

Wheel-based running was utilized to assess circadian activity. The mice were placed in individual cages with running wheels (Coulbourn Instruments, Allentown, PA, USA; ACT-551-MS-SS) in a different room with the same light and dark cycles and continuously monitored for 9 days (216 h) using the ClockLab Actiview Biological Rhythm Analysis Version 6 (Minimitter Co., Wilmette, IL, USA).

### 2.6. Brain Analysis

After the behavioral analyses were completed, the mice were euthanized in accordance with relevant guidelines, and their brains were extracted and allocated to either Western blots or thioflavin-S antibodies, chemicals, and reagents, as listed in Table 3.

#### 2.6.1. Western Blot Analyses for Protein Expression

The right hemispheres (n = 6 per group) (Table 4) were homogenized and run through the Western blotting procedure as previously described in Lippi et al., 2018 [19]. The samples were prepared using 40 µg of protein. The proteins of interest included pSer396, Tau5, pSer202, pThr231, and GFAP; the two loading controls were β-actin and GAPDH.

#### 2.6.2. Thioflavin-S for Tau Tangle Staining

Fresh-frozen brain tissue (Table 5) was sliced at 16µm using a Leica CM3050S cryostat. The coronal prefrontal (Bregma: 2.145 mm) and hippocampal sections (Bregma: −1.755 mm) were mounted onto slides and ran through the thioflavin-S staining procedure as previously described in Lippi et al., 2018 [19]. The slides were imaged under an Olympus BX51 fluorescence microscope equipped with a fluorescein isothiocyanate (FITC) cube and a mercury burner at a magnification of 20×; the green levels were at 1.26 and the exposure time was 0.28 s for all images. The prefrontal cortex, somatosensory cortex, dentate gyrus, and CA1 and CA3 regions of the hippocampus were examined using the ImageJ software version 1.46r (NIH, Bethesda, MD, USA) to automatically count and measure the tangles in each image. The parameters for the Analyze Particle features were set from 350 to infinity pixels and circularity was set from 0.10 to 1.00 for all images. 

### 2.7. Statistical Testing

All statistical analyses were conducted using the IBM SPSS Version 22, with significance being *p* < 0.05. Following the ANOVA/MANOVA analyses, pairwise comparisons were used to compare individual groups. The independent variables in each ANOVA consisted of genotype (WT or Tau) × injection (saline or plasma). Two 2 × 2 × 2 ANOVAs were conducted for (a) nesting scores at 2 and 12 h and (b) burrowing weight of remaining pea shingles in the burrow tubes at 2 and 12 h.

The open field test was analyzed using (i) percent time spent in surrounding areas; (ii) distance traveled; and (iii) velocity.

#### 2.7.1. Morris Water Maze

A 2 × 2 × 6 repeated measures ANOVA was performed on latency over the 6 days of testing. Additionally, 2 × 2 × 3 repeated measures ANOVAs were performed on percent time spent in the target quadrant, number of platform crosses on the probe trial days, and percent time spent near the walls (on days 2, 4, and 6). A 2 × 2 factorial ANOVA was performed for day 7.

#### 2.7.2. Circadian Rhythm

A 2 × 2 × 2 ANOVA was performed on the onset of activity between 9 PM and 10 PM, using wheel rotations as the dependent variable.

#### 2.7.3. Western Blot

A 2 × 2 ANOVA was performed to assess the differences in relative density values. The proteins used were Tau5, pSer396, and GFAP. Three isoforms of GFAP (GFAP-a, GFAP-ΔEx7, and GFAP-Δ164) were analyzed separately. The proteins of interest were normalized to their corresponding loading controls. Subsequently, pThr231 and pSer 202 were analyzed for the Tau mice with and without plasma injection, using *t*-tests.

#### 2.7.4. Thioflavin-S

A 2 (injection) × 5 (brain area) ANOVA was performed for the number of tangles and the size of tangles.

## 3. Results

The plotted values for all figures are represented as mean ± SEM.

### 3.1. Brain Analysis

#### 3.1.1. Thioflavin-S in Tangles

The mice injected with plasma had fewer tangles in the cortex (F(1,8) = 6.669, *p* = 0.032, $\eta_{p}^{2}$ = 0.455) (Figure 2 and Figure 3) and in the CA1 region (F(1,8) = 5.717, *p* = 0.044, $\eta_{p}^{2}$ = 0.417) (Figure 2A) compared to the mice injected with saline. There was a significant main effect for injection in the cortex area: the mice injected with plasma had smaller tangles in the somatosensory cortex compared to the mice injected with saline (F(1,8) = 26.214, *p* = 0.001, $\eta_{p}^{2}$ = 0.766) (Figure 2B).

#### 3.1.2. Western Blot

(i)Total Tau

The Tau mice had more tau expression in the brain than the WT mice (F(1,12) = 230.677, *p* < 0.001, $\eta_{p}^{2}$ = 0.920) (Figure 4A). There was no main effect of injection and no interaction between genotype and injection.

(ii)Phosphorylated Tau

The Tau mice had more phosphorylated tau expression in the brain than the WT mice for pSer396 (F(1,12) = 22.300, *p* < 0.001, $\eta_{p}^{2}$ = 0.527) (Figure 4B). There was an interaction effect between genotype and injection (F(1,12) = 4.483, *p* = 0.047, $\eta_{p}^{2}$ = 0.183). The Tau mice given plasma had less phosphorylated tau expression compared to the Tau mice given saline (*p* < 0.05). For pThr231, t(1,6) = 5.548, with *p* = 0.057. For pSer 202, t(1,6) = 1.7, NS.

(iii)GFAP

The Tau mice had more total GFAP expression in the brain compared to the WT mice (F(1,12) = 10.100, *p* = 0.005, $\eta_{p}^{2}$ partial = 0.336) (Figure 5). The Tau mice had more GFAP-α (F(1,12) = 13.587, *p* = 0.001, $\eta_{p}^{2}$ partial = 0.405)*,* GFAP-ΔEx7 (*F*(1,12) = 11.399, *p* = 0.003, $\eta_{p}^{2}$ partial = 0.363), and GFAP-Δ164 (*F*(1,12) = 11.112, *p* = 0.003, $\eta_{p}^{2}$ partial = 0.357) expressions than the WT mice. There was no main effect of injection and no interaction between genotype and injection for any of the GFAP analyses.

### 3.2. Behavioral Analysis

#### 3.2.1. Nesting

The wildtype mice ($\overset{-}{x}$ = 3.68, SE = 0.19) built significantly better nests than the Tau mice ($\overset{-}{x}$ = 2.53, SE = 0.19) (F(1,54) = 19.46, *p* < 0.001, $\eta_{p}^{2}$ = 0.265).

#### 3.2.2. Burrowing

The wildtype mice ($\overset{-}{x}$ = 154.98, SE = 10.59) burrowed significantly more than the Tau mice ($\overset{-}{x}$ = 59.141, SE = 10.54) (*F*(1,54) = 41.171, *p* < 0.001, $\eta_{p}^{2}$ a = 0.433).

#### 3.2.3. Open Field

(i)Time spent in the surrounding area

The Tau mice spent significantly more time in the surrounding area compared to the WT mice (*F*(1,54) = 6.339, *p* = 0.015, $\eta_{p}^{2}$ a = 0.105). 

(ii)Distance Traveled

The Tau mice traveled significantly further compared to the WT mice (F(1,54) = 4.083, *p* = 0.048, $\eta_{p}^{2}$ a = 0.070). 

(iii)Velocity

There were no significant effects on velocity.

#### 3.2.4. Morris Water Maze

(i)Latency to Platform

The Tau mice had significantly longer latencies to reach the platform compared to the WT mice (*F*(1,54) = 44.233, *p* < 0.001, $\eta_{p}^{2}$ a = 0.450) (Figure 6A). A significant within-subject effect of day (*F*(1,54) = 13.351, *p* < 0.001, $\eta_{p}^{2}$ a = 0.198) showed that latency to reach the platform decreased across days.

(ii)Target Quadrant

The Tau mice spent significantly less time in the target quadrant than the WT mice (F(1,54) = 4.786, *p* = 0.033, $\eta_{p}^{2}$ a = 0.081).

(iii)Platform Crosses

The Tau mice had significantly fewer platform crosses compared to the WT mice (F(1,54) = 36.288, *p* < 0.001, $\eta_{p}^{2}$ a = 0.402) (Figure 6B).

(iv)Thigmotaxicity

The Tau mice spent significantly more time in the outer 10% of the pool compared to the WT mice (*F*(1,54) = 33.473, *p* < 0.001, $\eta_{p}^{2}$ a = 0.383)(Figure 6C). A significant effect of day (F(1,54) = 7.807, *p* = 0.007, $\eta_{p}^{2}$ a = 0.126) showed that there was less time spent in the outer 10% of the pool as the days progressed.

(v)Swim Speed

On day 8, the visual platform day, the WT mice ($\overset{-}{x}$ = 30.93, SE = 14.87) were significantly faster than the Tau mice ($\overset{-}{x}$ = 41.90, SE = 16.96) (*F*(1,54) = 6.303, *p* = 0.015, $\eta_{p}^{2}$ a = 0.105).

#### 3.2.5. Circadian Rhythm

The Tau mice had lower activity than the WT mice during the onset of activity from 9 pm to 10 pm (F(1,48) = 4.475, *p* = 0.040, $\eta_{p}^{2}$ = 0.085).

## 4. Discussion

The results from the current study showed significant improvement with plasma injections in the brains of Tau mice, indicating plasma had an effect at a cellular level. Two bands of β-actin loading control were seen (Figure 4B), indicating tau induced actin degeneration [20,21]. The Tau mice injected with plasma had significantly fewer tangles in the somatosensory cortex and CA1 region of the hippocampus compared to the tau mice injected with saline (Figure 3), as well as significantly smaller tangles in the somatosensory cortex. As expected, the WT mice had little to no expression of total or phosphorylated tau. Total tau was not significantly different in the Tau mice injected with saline or plasma. The initial antibody used for hyperphosphorylated tau was pSer396, an isoform heavily expressed in the CA1 region of the hippocampus [7], Tau phosphorylated at Ser396 was significantly lower and tau phosphorylation at Thr231 was also reduced (*p* < 0.06) in the Tau mice given plasma compared to those injected with saline, suggesting that plasma might prevent the formation of new NFTs. Interestingly, tau phosphorylation at Ser202 was not affected by plasma injection. This might be because pSer202 hyperphosphorylated at an earlier stage of tauopathy [22]. Pedemonte et al. gave plasma at four months to Tau mouse, which led to decreased pSer202 in the plasma-treated mice and some significant improvements in behavior [23]. The later blood plasma is given, the more difficult it is to clear out tau tangles or prevent the formation of more tangles, which in turn makes it difficult to alleviate the behavioral symptoms caused by tau.

Although there was a reduction in the number and size of tangles, this did not alleviate the behavioral symptoms. As expected, the Tau mice did display spatial memory impairments in the MWM. However, plasma administration did not significantly change the performance of either the Tg or WT mice. The Tau mice had higher thigmotaxicity during the MWM compared to the WT mice, and in the open field test, the Tau mice spent more time in the surrounding area compared to the WT mice, indicating higher anxiety. Two forms of activities of daily living (ADL), nesting and burrowing, were tested; the Tau mice built worse nests and burrowed less than the WT mice, but there was no plasma effect. AD patients suffer from disturbed circadian rhythm and irregular sleep–wake cycles [24]. In the current study, the Tau mice were less active during the first two hours in the dark cycle compared to the WT mice.

Overall, the Tau mice performed worse than the WT mice, but plasma had no significant effect. Plasma might have been ineffective due to multiple reasons. For one, the injections might have been given too late. The mice were eight months of age when given young plasma; NFTs had developed in the hippocampus and 43–60% of neuronal loss in the hippocampus was present, which were used to assess whether blood plasma transfusions could function as a treatment after the progression of the disease. If plasma injections were given before eight months, it might have had a more significant effect.

Alternatively, a greater amount of young plasma might have been needed. Zhao et al. (2020) administered young plasma to 3xTg mice at eight months of age for eight weeks [25]. This was a longer dosing period and a higher amount than the dose given by Middeldorp et al. (2016) [12], Villeda et al. (2014) [11], and in the current study. Zhao et al. found improvements in behavior as well as reductions in phosphorylated tau and amyloid plaques; NFTs were not examined [25]. Zhao et al. indicated that tangles did not develop in the 3xTg mice until 12 months of age after young plasma was given, so they were able to prevent the pathology [25]. In the case of the rTg4510 Tau mouse model, tangles have been seen as early as four months, although Gamache et al. (2019) have suggested that this may be due in part to the disruption of an endogenous mouse gene [26]. Thus, in the present study, tangles were well established before any young plasma was administered. 

Another limitation might be the rTg4510 mouse model itself. In this specific regulatable model, the P301L tau mutation and CaMKIIa promoter system result in progressive neurofibrillary tangle pathology within the forebrain and memory deficits over time, which can be lessened when given doxycycline. However, as Gamache et al. (2019) reported, this phenotype may not be caused solely by expressed tau [26]. Gamache et al. (2019) reported that, in this model, disruptions in genes, including fibroblast growth factor 14 (Fgf14) caused by the insertion of the transgene itself may be responsible for the particular phenotypes reported [26]. While this model still allows researchers to assess tangle pathology and behavioral deficits as a function of age, when applying an intervention aimed at alleviating behavioral or biochemical deficits, researchers must consider other factors that may be causing the deficits.

It is not clear why giving young blood causes significant behavioral and/or brain changes in both WT and Tg mice. The positive factors that have been proposed include a reduction in abnormal ERK signaling [22] associated with the increased phosphorylation seen in AD. Growth differentiation factor 11 (GDF11) has also been identified as a rejuvenating factor in young blood and has been shown to enhance neurogenesis and cerebral blood flow in aged mice [27,28]. Previous studies in 2014, with one of them conducted by Katsimpardi et al., that used parabiotic mice described GDF11 as an anti-aging factor [28]. They were able to show that GDF11 improved cerebral vasculature and enhanced neurogenesis. However, there has been a debate as researchers disagree on the selectivity of the tests to measure GDF11 and on the activity of GDF11 from commercial sources. The relationship between GDF11 and aging is still being researched. Middeldorp et al. (2016) showed that ERK signaling was abnormal in Alzheimer’s disease mouse models [12]. Through heterochronic parabiosis, they were able to improve the signaling by restoring synaptophysin and calbindin protein levels, which are important for synaptic transmission based on calcium ion binding. Thus, they concluded that plasma reduced activation of the ERK pathway. Although there is a debate on the involvement of ERK and tauopathy, some researchers found that inhibiting ERK1/2 did not alter tau phosphorylation neither in vivo nor in vitro [29], while others found that a specific ERK inhibitor reduced and prevented tau aggregations in living cells [30]).

Other possibilities include *Iba1*, a microglial marker which Zhao et al. (2020) showed was reduced in the 3xTg mouse model, indicating reduced inflammation [25]. Zhao et al. (2020) also observed increases in GFAP. GFAP in this study was higher in the Tau mice compared to the WT mice, as expected, indicating an increase in glial activation, which may be a factor in impaired synaptic plasticity [31]. Increase in GFAP is also associated with increased neuronal death [32], mirroring the pathology in AD. Unfortunately, plasma injections did not appear to have an effect on individual GFAP isoforms, nor an effect on total GFAP expression. GFAP-α is the primary isoform abundantly found in mouse brains, both in wildtype and transgenic mice. In AD, mouse models have been shown to increase in expression in the cortex and hippocampus with age [33]. GFAP-ΔEx7 is an unestablished Isoform in mouse brains, and its transcription expression is relatively low and requires more research [33]. GFAP-Δ164 has been identified in pyramidal neurons of the hippocampus in Alzheimer’s patients [34]. This explains why, in the Western blot images, it is only seen in the Tau mice, since these mice carry the gene for this GFAP isoform to be expressed in the brain. Therefore, higher expression of GFAP-Δ164 is associated with behavioral deficits caused by tau tangles in the brain.

Although Middeldorp et al. did not observed changes in inflammation, Zhao et al. suggested that reduction in neuroinflammation and Aβ deposition contributed to the reduction in tau phosphorylation. Giving old blood to young mice led to impairments in behavior [11]. Thus, there is a question as to whether it is the contribution of the positive attributes of young blood or the removal of negative factors that is responsible for the changes seen. These two possibilities may both contribute.

## 5. Conclusions

In summary, this study showed that young plasma, which was given after tangles had developed, reduced the number of tangles but did not change behavior, thereby providing additional findings about the interaction between young plasma injections and NFTs in the brain. Future studies should further research the effects of injecting AD mice at a younger age. It is possible that plasma transfusion may be more effective if used as a preventative measure rather than as a treatment.

## Figures and Tables

**Figure 1 brainsci-13-00841-f001:**
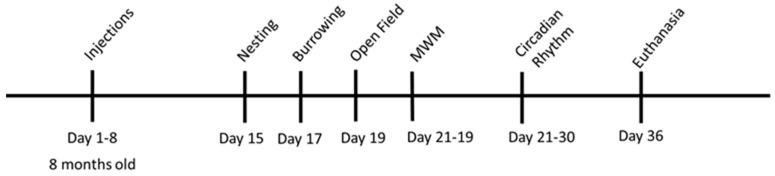
Timeline of behavioral test.

**Figure 2 brainsci-13-00841-f002:**
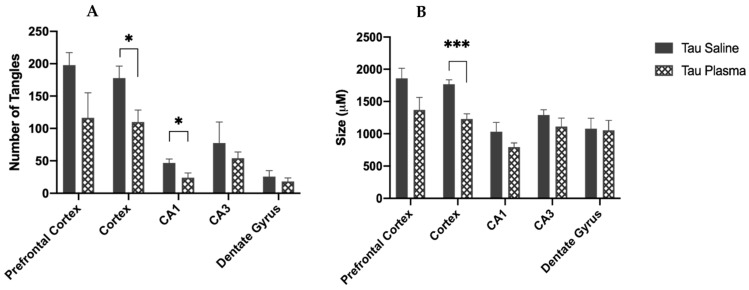
(**A**) Number of tau tangles in the brain of Tau mice. Tau mice injected with saline have more tangles in the somatosensory cortex (*p* = 0.032) and CA1 region of the hippocampus (*p* = 0.044) compared to the Tau mice injected with plasma. (* = *p* < 0.05). (**B**) Size of tangles in the brain of Tau mice. Tau mice injected with saline have larger tangles in the somatosensory cortex (*p* = 0.001) compared to the Tau mice injected with plasma. (*** = *p* < 0.001).

**Figure 3 brainsci-13-00841-f003:**
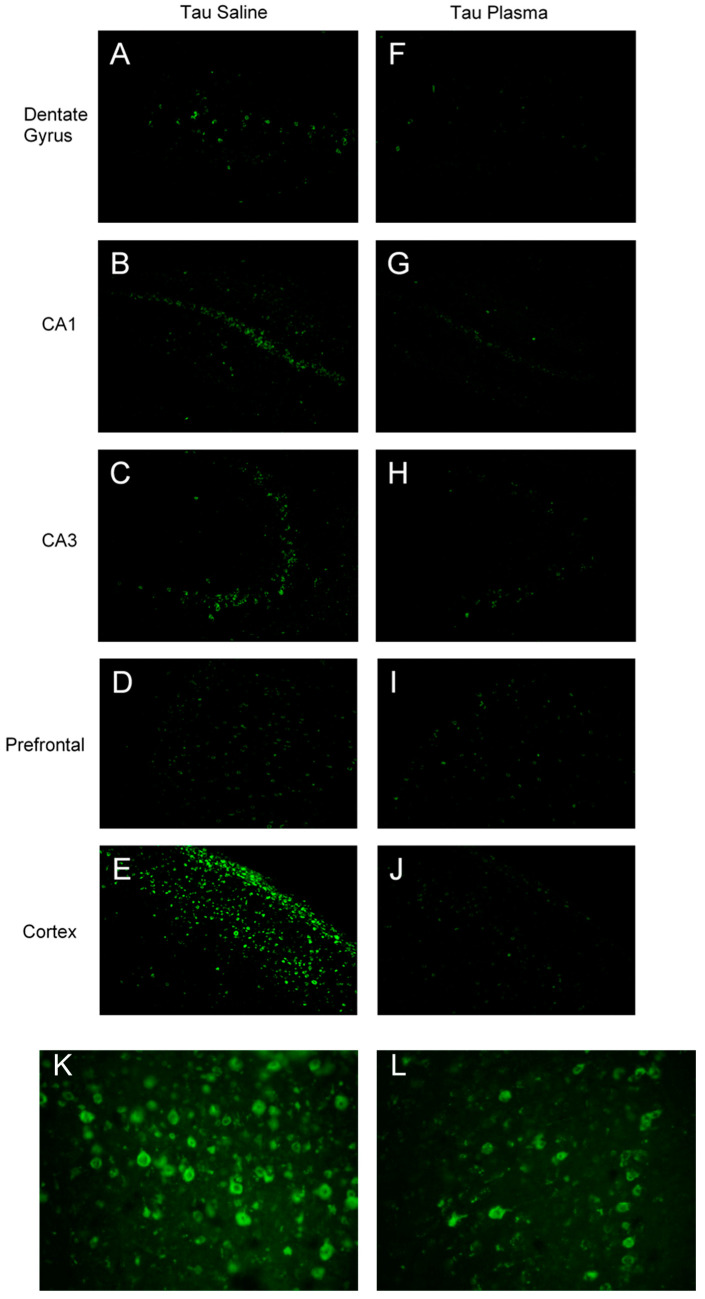
Thioflavin-S tau tangle microscope images at 20× magnification: the green levels were set to 1.26, and the exposure time of 0.28 s was consistent for all images. The images were captured from similar brain areas to show comparison between plasma-injected mice and saline-injected mice. (**A**–**E**) images are from Tau mice given saline injections. (**F**–**J**) images are from Tau mice given plasma injections. (**K**,**L**) are higher-magnification images to show the pathology of tau tangles from the prefrontal cortex of Tau mice.

**Figure 4 brainsci-13-00841-f004:**
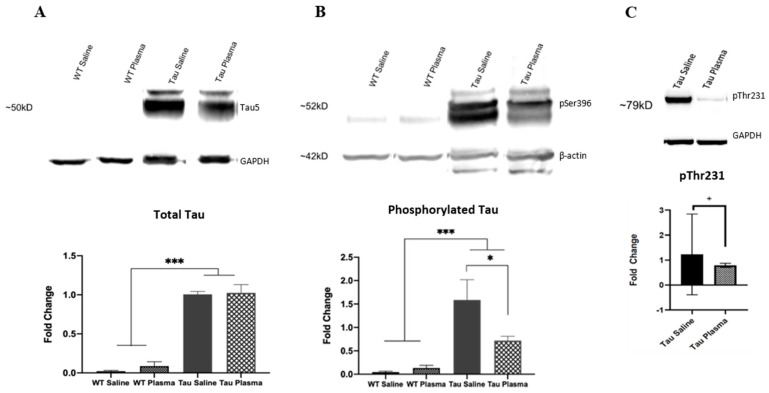
(**A**) Tau5 Western blot analysis. Tau mice, regardless of injection, have a greater expression of tau than WT mice. (*** = *p* < 0.001). Tau 5 ~50kD; loading control GAPDH ~35kD. (**B**) pSer396 Western blot analysis. Overall, Tau mice have a higher expression of phosphorylated tau than WT mice (*p* = 0.001). Additionally, Tau mice injected with saline have a higher expression of phosphorylated tau than Tau mice injected with plasma (*p* = 0.047). (* = *p* < 0.05, *** = *p* < 0.001). pSer396 ~52kD; loading control β-actin ~42kD (double bands indicate actin degradation). (**C**) pThr231 Western blot analysis. Tau mice injected with plasma have a lower expression of phosphorylated tau than Tau mice injected with saline (F(1,6) = 5.548, *p* = 0.057). (+ = *p* < 0.05). pThr231 ~79 kD; loading control GAPDH ~35kD.

**Figure 5 brainsci-13-00841-f005:**
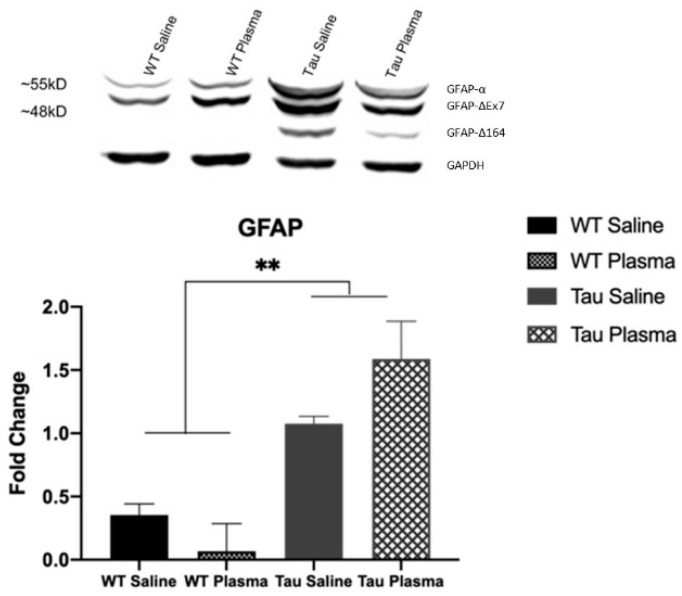
GFAP Western blot analysis. Tau mice have a higher expression of astrocyte activity than WT mice. (** = *p* < 0.01). GFAP-α ~55kD; GFAP-ΔEx7 ~48 kD; GFAP-Δ164 between ~48 kD and ~35 kD; loading control GAPDH ~35 kD.

**Figure 6 brainsci-13-00841-f006:**
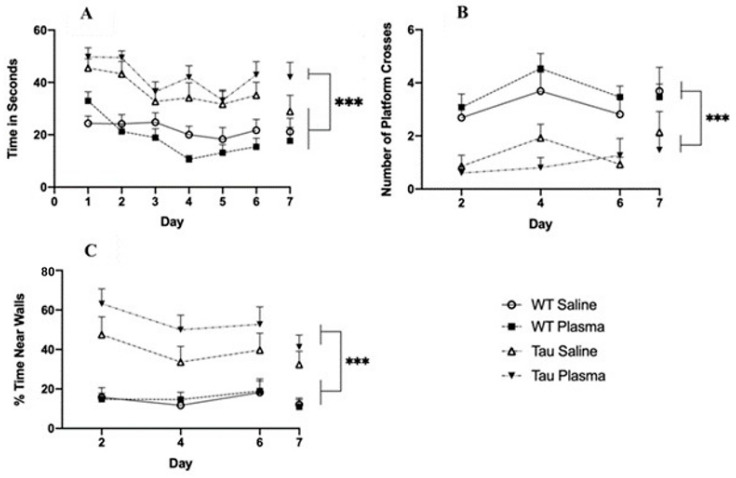
(**A**) MWM latency to find platform. Results show that Tau mice have a significantly longer latency to reach the platform compared to WT mice, regardless of injection (*p* < 0.001). There is a significant effect of day (*p* < 0.001); the mice reached the platform faster on day 6 than on days 1 (*p* < 0.001), 2 (*p* = 0.019), and 5 (*p* < 0.015). (*** = *p* < 0.001). (**B**) MWM platform crosses. Tau mice make significantly fewer platform crosses compared to WT mice regardless of injection (*p* < 0.001). Day 7 was a probe day and was, thus, separated from the other days. (*** = *p* < 0.001). (**C**) MWM Thigmotaxicity. Tau mice spend significantly more time in the outer 10% of the pool compared to WT mice. There is a significant effect of day (*p* < 0.001); the mice spent significantly less time near the walls on day 7 than on days 2 and 6 (*p* < 0.0001). (*** = *p* < 0.001).

**Table 1 brainsci-13-00841-t001:** Behavioral groups.

	Tau		WT	
	Male	Female	Male	Female
Plasma	9	6	7	6
Total	15			13
Saline	7	7	11	5
Total	14			16

Total sample size is 58.

**Table 2 brainsci-13-00841-t002:** Morris water maze protocol.

Day	Number of Trials	Platform Location
1	1, 2, 3	Stationary and submerged *
2	1, 2, 3 (Probe)	1 and 2: Stationary and submerged; 3: Atlantis **
3	1, 2, 3	Stationary and submerged
4	1, 2, 3 (Probe)	1 and 2: Stationary and submerged; 3: Atlantis
5	1, 2, 3	Stationary and submerged
6	1, 2, 3 (Probe)	1 and 2: Stationary and submerged; 3: Atlantis
7	1 (Probe)	Atlantis
8	1, 2	Visible (lighthouse) ***

All trials had a duration of 60 s. * Stationary and submerged: the platform was 5 mm below the water’s surface, and the mice were able to stand on the platform. ** Atlantis: the platform was completely lowered, and the mice were unable to stand on the platform. *** Visible: the platform was 5 mm above the water’s surface and affixed with a distinct “lighthouse” cue to attract the mice to the platform.

**Table 3 brainsci-13-00841-t003:** Antibodies, chemicals, and reagents.

Substance	Manufacturer/Product	Useage
pSer396 antibody	Abcam (Cambridge, UK): ab109390	Western blot (1:10,000) phosphorylated tau
Tau5 antibody	ThermoFisher Scientific (Waltham, MA, USA): AHB0042	Western blot (1:1000) total tau
GFAP antibody	ThermoFisher Scientific (Waltham, MA, USA): MA5-12023	Western blot (1:10,000) astrocytes
β-actin antibody	ThermoFisher Scientific (Waltham, MA, USA): MA5-15739	Western blot (1:10,000) loading control
GAPDH antibody	ThermoFisher Scientific (Waltham, MA, USA): MA5-15738	Western blot (1:10,000) loading control
HRP-conjugated secondary antibody	ThermoFisher Scientific (Waltham, MA, USA): A27036, A28177	Western blot
RIPA buffer	ThermoFisher Scientific (Waltham, MA, USA): 89901	Western blot brain homogenate
Halt Protease and Phosphatase Inhibitor Cocktail	ThermoFisher Scientific (Waltham, MA, USA): 78446	Western blot brain homogenate
NuPAGE 4–12% Bis-Tris gels	Invitrogen (Waltham, MA, USA): NP0321BOX	Western blot
NuPAGE MOPS running buffer	Invitrogen (Waltham, MA, USA): NP0001	Western blot
West pico chemiluminescent substrate (SuperSignal)	ThermoFisher Scientific (Waltham, MA, USA): 34579	Western blot
Potassium permanganate	Sigma Aldrich (St. Louis, MO, USA): 223468	Thioflavin-S Tau stain
Sodium borohydride caplets	Sigma Aldrich (St. Louis, MO, USA): 452890	Thioflavin-S Tau stain
Thioflavine S	Sigma Aldrich (St. Louis, MO, USA): T1892	Thioflavin-S Tau stain

**Table 4 brainsci-13-00841-t004:** Western blot groups.

	Tau		WT	
	Male	Female	Male	Female
Plasma	4	2	3	3
Total	6			6
Saline	3	3	4	2
Total	6			6

Total sample size is 24.

**Table 5 brainsci-13-00841-t005:** Thioflavin-s groups.

	Tau		WT	
	Male	Female	Male	Female
Plasma	3	2	1	1
Total	5			2
Saline	2	3	1	1
Total	5			2

Total sample size is 14.

## Data Availability

Not applicable.

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
