# Peer review of "Transfusion with Blood Plasma from Young Mice Affects rTg4510 Transgenic Tau Mice Modeling of Alzheimer’s Disease"

_brainsci, 2023, doi:10.3390/brainsci13060841_

Round 1

Reviewer 1 Report

The investigators studied the effect of young plasma to 8-month old tau mice and observed reduction in amount of hyperphosphorylated tau but no effect on behavioural test results. The findings will stimulate momentum for further study hopefully with results translated to benefit of dementia patients.

Mice injected with young plasma at different and earlier ages are much needed for better understanding.

I suggest to improve the manuscript by strengthening the discussion on potential mechanisms underlying neuroprotective effects of young plasma, for example more details concerning effect on ERK signaling and role of GDF11.

satisfactory, some typos

Author Response

We have included the following text in the manuscript:

Studies in 2014, one of them being Katsimpardi et al (2018) that used parabiotic mice described GDF11 as an anti-aging factor [18]. However, there has been debate as researchers disagree on the selectivity of the tests to measure GDF11 and on the activity of GDF11 from commercial sources. The relationship between GDF11 and aging is still being researched. Middeldorp et al (2016) showed that ERK signaling is abnormal in Alzheimer’s Disease mouse models [6]. Through heterochronic parabiosis they were able to improve the signaling by restoring synaptophysin and calbindin protein levels important for synaptic transmission based on calcium ion binding. Thus concluding that plasma reduced activation of the ERK pathway. Although there is debate on the involvement of ERK and tauopathy, some have found that inhibiting ERK1/2 did not alter tau phosphorylation in vivo nor in vitro (Noel et al 2015), while others have found that a specific ERK inhibitor reduced and prevented tau aggregations in living cells (Siano et al 2019).

Reviewer 2 Report

In this study, the authors showed that plasma transfusion from young donor mice improved tau pathology in 8-month-old rTg4510 mice expressing human tau, whereas no improvements in behavior were observed. Although the results are generally interesting, I have following concerns.

1. There is no statement about whether the animal experiments in this study was approved by the institutional animal committee. 

2. In Page 8, line 205, the authors described that “Tau mice had more phosphorylated tau expression in the brain than WT mice for pSer396 (Fig. 4B)”. However, in Fig 4B, the graph does not show pSer396 levels, but Total Tau levels.

3. In Fig. 4 and 5, it is not clear which band shows Tau, GFAP, beta-Actin, and GAPDH, respectively.

4. In the References, the details of the reference No. 13 are missing.

Author Response

1. We added the following text in the manuscript: The study described in this paper was approved by the George Mason University Institutional Animal Care and Use Committee under protocol #0390 entitled, “Effects of Blood Plasma Transfusion from Young Mice to Aged h-tau Mice Modelling Alzheimer’s Disease”. The IRBNet package number was 1195958-1. Procedures described herein were in accordance with AVMA and The Guide for the Care and Use of Laboratory Animals, as well as AAALAC guidelines.

2. The figures have been updated and re-arranged to show the correct graph (please see fig. 4 in attachment or revised manuscript)

3. Figures were updated to include antibody labels next to each band (please see fig. 5 in attachment or revised manuscript)

4. Citation was added in manuscript: Pedemonte et al., (2021) The Effects of Young Blood Plasma Transfusions on Older hTau Mice Modeling Alzheimer’s Disease. George Mason University. http://hdl.handle.net/1920/13223

Reviewer 3 Report

In the current study, authors studied the impact of young plasma in Tau Tg mice. Authors  transfused from 2-3-month young wild-type mice (WT) to 8-month-old rTg4510 mice expressing human tau (Tau). One week after transfusions cognitive behavior and tau pathology were examined. They found that Tau mice injected with plasma had lower expression of phosphorylated tau (ptau) in the brain accompanied by fewer tau tangles in the cortex and CA1 region of the hippocampus, but no behavioral changes. Interesting preliminary observation – it may be worth looking at activities such as inflammation, synapses and mitochondria. Concerns - literature survey is not adequate - refer Rawat et al 2022, IJMS

Author Response

We agree with the statement at looking at inflammation, synapses, and mitochondria. At the time this study was conducted we highly considered doing experiments with synapses by looking at long term potentiation. This required collaboration with another lab at our institute, unfortunately due to logistics and lack of resources we had to drop that part of the study.

We’ve included additional citations to our paper and references. Some of these we previously removed because other reviewers requested we trim our paper down. Additional text and references are highlighted in the paper (please see revised manuscript)